Research




**Subject Areas:**
bioinformatics/genomics/immunology

**Keywords:**
vaccinology, *Mycoplasma pneumoniae*, bioinformatics, genomic, molecular docking, pneumonia

**Author for correspondence:**
Siomar de Castro Soares
e-mail: siomars@gmail.com

[†]These authors contributed equally to this study.



# Reverse vaccinology and subtractive genomics reveal new therapeutic targets against *Mycoplasma pneumoniae*: a causative agent of pneumonia

Thaís Cristina Vilela Rodrigues[1,†], Arun Kumar Jaiswal[1,2,†], Alissa de Sarom[1], Letícia de Castro Oliveira[1,2], Carlo José Freire Oliveira[1], Preetam Ghosh[3], Sandeep Tiwari[2], Fábio Malcher Miranda[2], Leandro de Jesus Benevides[4], Vasco Ariston de Carvalho Azevedo[2] and Siomar de Castro Soares[1]

[1]Department of Microbiology, Immunology and Parasitology, Federal University of Triângulo Mineiro, Minas Gerais, Brazil
[2]Department of Genetics, Ecology and Evolution, Federal University of Minas Gerais, Minas Gerais, Brazil
[3]Department of Computer Science, Virginia Commonwealth University, Richmond, VA 23284, USA
[4]Bioinformatics Laboratory - LABINFO, National Laboratory of Scientific Computation - LNCC/MCTI, Rio de Janeiro, Brazil

TCVR, 0000-0002-2048-5522; AdS, 0000-0002-3146-7784; LdCO, 0000-0002-2036-4456; PG, 0000-0003-3880-5886; ST, 0000-0002-8554-1660; FMM, 0000-0002-6823-5995

Pneumonia is an infectious disease caused by bacteria, viruses or fungi that results in millions of deaths globally. Despite the existence of prophylactic methods against some of the major pathogens of the disease, there is no efficient prophylaxis against atypical agents such as *Mycoplasma pneumoniae*, a bacterium associated with cases of community-acquired pneumonia. Because of the morphological peculiarity of *M. pneumoniae*, which leads to an increased resistance to antibiotics, studies that prospectively investigate the development of vaccines and drug targets appear to be one of the best ways forward. Hence, in this paper, bioinformatics tools were used

for vaccine and pharmacological prediction. We conducted comparative genomic analysis on the genomes of 88 *M. pneumoniae* strains, as opposed to a reverse vaccinology analysis, in relation to the capacity of *M. pneumoniae* proteins to bind to the major histocompatibility complex, revealing seven targets with immunogenic potential. Predictive cytoplasmic proteins were tested as potential drug targets by studying their structures in relation to other proteins, metabolic pathways and molecular anchorage, which identified five possible drug targets. These findings are a valuable addition to the development of vaccines and the selection of new *in vivo* drug targets that may contribute to further elucidating the molecular basis of *M. pneumoniae*–host interactions.

# 1. Introduction

The genus *Mycoplasma* belongs to the class Mollicutes, which are bacteria without a cell wall. Through comparative genomics and phylogenetic analysis, it has been suggested that these bacteria probably originated from Gram-positive ancestors [1–3]. Bacteria of the genus *Mycoplasma* are the smallest known microorganisms in cell and genomic size with a capacity for self-replication [4]. Owing to the fact that the *Mycoplasma* genome comprises fewer than 1000 genes, its metabolic capacities are reduced, thus it requires specific cellular compounds for its survival. For their growth in culture media, these microorganisms require singular components such as sterols that protect them against their osmotic fragility. The absence of a cell wall in the genus *Mycoplasma* makes it difficult to classify these microorganisms as cocci or bacilli; furthermore, this characteristic confers a natural resistance to β-lactams and impairs Gram staining [5,6]. Among the 16 *Mycoplasma* species that infect humans, six of them are pathogenic and, of these, *Mycoplasma pneumoniae* is the one with the highest clinical significance [7,8].

*Mycoplasma pneumoniae*, the main causative agent of community-acquired pneumonia (CAP), has a genome of approximately 800 000 base pairs that encodes approximately 700 different proteins [9]. Studies have demonstrated a prevalence rate of CAP for *M. pneumoniae* that can reach 40% among confirmed cases [10]; however, it is still extremely difficult to make a specific diagnosis in asymptomatic cases. CAP is an acute lung infection responsible for high morbidity and mortality rates and tends to be contracted by individuals outside the healthcare system [11]. In most cases, the pathogen is not identified and the diagnosis is sometimes based only on clinical signs; consequently, the treatment is not specific, and therapeutic intervention may compromise the patient's life and even contribute to the development of antibiotic-resistant bacteria [12].

Because of the absence of the cell wall, infections caused by *M. pneumoniae* are treated using antibiotics from the macrolide, tetracycline and quinolone classes [13,14]. It is known that a mutation in the 23S rRNA gene conferred resistance to macrolides in several strains, reducing treatment options to only two classes of antibiotics. Asia is the continent with the highest rate of resistance, where about 90% of the strains of this bacterium have already demonstrated this mutation. This high rate was due to the frequent and indiscriminate antibiotic use in this region. Cases of resistance in Europe and America have also been reported and the proportion of resistant strains has been growing progressively. Thus, there is a need to promote not only restrictions for the use of macrolides, but also advancement of research for the development of new drugs and prophylactic methods [15–17]. There are currently vaccines against *Streptococcus pneumoniae* and *Haemophilus influenzae*, the most common causative agents of pneumonia. On the other hand, for atypical pathogens that cause this disease, such as *M. pneumoniae* and *Chlamydia pneumoniae*, studies are still needed to achieve the same level of progress [18]. Such studies can be extremely important since, according to the World Health Organization [19], pneumonia vaccines prevent the deaths of almost 2 million children per year [20].

With the advancement of genome sequencing technologies, the number of microorganism species with completed genome sequences has increased rapidly, adding thousands of new sequenced genomes to databases and providing material for numerous types of studies, including the prediction of new drug targets and vaccines, through approaches such as comparative and subtractive genomics [21–23]. Another approach is reverse vaccinology, which is also based on the use of the genomic sequence of a given microorganism for *in silico* analysis. Using this bioinformatics tool, it is possible to investigate, in several ways, all of the proteins that can be produced by the bacterium and evaluate their ability to induce an adaptive immune response or to bind to drugs [24,25]. Reverse vaccinology optimizes the prediction of drug and vaccine targets, especially for microorganisms that are difficult to grow in the laboratory, such as intracellular bacteria including *M. pneumoniae*. In addition, reverse vaccinology allows the simultaneous analysis of targets in multiple genomes, which is important for the predicted targets to achieve greater

coverage among lineages [26]. The use of reverse vaccinology gained prominence in 1995 with the publication of the complete genome of the bacterium *H. influenzae* [27]. Subsequently, this methodology was used for the screening of antigens to prepare vaccines against *Neisseria meningitidis* [28,29], *Acinetobacter baumannii* [30], *Streptococcus agalactiae* [31], human cytomegalovirus, respiratory syncytial virus, human immunodeficiency virus, influenza and dengue virus [32,33], among other pathogenic agents.

Given the relevance of the problem, the objective of this work was to predict drug and vaccine targets through bioinformatics tools, by selecting those that act against all 88 *M. pneumoniae* strains whose genomes are already deposited in GenBank. This study will facilitate future *in vitro* and *in vivo* tests for the production of drugs and prophylactic targets against a species of bacteria with high clinical relevance for CAP, especially among children.

# 2. Methodology

## 2.1. Genomes

The 88 genomes of *M. pneumoniae* strains available in the GenBank database were downloaded through the National Center for Biotechnology Information (NCBI) for the bioinformatics analysis. For this, both the complete and incomplete downloaded genomes were converted to the FASTA format.

## 2.2. Identification of conserved proteins of *M. pneumoniae* and subtractive genomics

The FASTA format files containing the amino acid sequences were submitted to the software OrthoFinder under its default parameters. The algorithm developed for this software performs calculations based on searches through BLAST and the MCL clustering algorithm to identify the regions of homology, thereby generating the orthogroups with the protein sequences. Subsequently, in-house scripts were employed to classify genes into three groups: core genes, which represent those present in all studied strains; shared genes, which are present in some, but not all, strains; and the singletons, which are strain-specific genes present in only one strain [34]. With the amino acid sequences (faa), a BLASTp was performed against the proteins of the human genome, also using the OrthoFinder, to identify the proteins belonging to the bacterium *M. pneumoniae* that have no homology with those from the host. This stage is called subtractive genomics and was essential to avoid the selection of drug targets or vaccines without protective effect or even those that may cause autoimmunity [35].

## 2.3. Characterization and prediction of the subcellular location of proteins of *M. pneumoniae*

To verify the importance of each of the identified proteins, we used the Database of Essential Genes (DEG), which includes all essential bacterial and eukaryotic gene records [36]. This online platform contains information on genes from bacteria, archaea and eukaryotes, responsible for the production of several proteins, as well as data from non-coding RNAs (http://www.essentialgene.org/). Only proteins considered essential to the microorganism were used in the prediction of candidate vaccine antigens and drug targets. SurfG+ is a software that predicts the subcellular localization of the proteins of interest. The prediction consists of identifying peptide signal, retention signals, transmembrane helices and secretion pathways to classify proteins as secreted, PSE (putatively exposed to the surface) and membrane proteins. Among the identified proteins, the cytoplasmic proteins were subjected to an analysis for potential drug targets, because of their involvement in the basic survival processes of the organism, while proteins characterized as membrane, PSE and secreted were directed to reverse vaccinology analysis, since they are the first proteins to come into contact with the immune response of the host [37].

## 2.4. Selection of drug targets and druggability analysis

The MHOLline program was used to model three-dimensional (3D) cytoplasmic proteins. This software combines other programs such as HMMTOP, BLAST, BATS, MODELLER and PROCHECK to analyse and classify potential drug targets according to their structural quality. BLAST performs a random search against protein databases (PDBs) and provides three-dimensional structures of the targets. The BATS (Blast Automatic Targeting for Structures) program selected the proteins in which the comparative modelling technique was applied and rated the models in seven groups according to quality (from 'very high' to 'very low'). Three-dimensional models and global alignment were

produced by the MODELLER program and evaluated for stereochemical quality through PROCHECK. To complement the process, transmembrane helix topology studies were performed by the software HMMTOP. The BATS program organized the BLAST output files into four groups—G0, G1, G2 and G3—following the criteria: G0, non-aligned sequence; G1, $E > 10 \times 10^{-5}$ or identity < 15%; G2, $E \leq 10 \times 10^{-5}$, identity ≥ 25% and length variation index (LVI) ≤ 0.7; G3, $E \leq 10 \times 10^{-5}$, identity ≤ 15% to <25% or LVI > 0.7 [38]. The 3D structures with identity < 25%, corresponding to groups G1 and G3, did not fit into the comparative modelling technique of the MHOLline program, and, thus, only the G2 group sequences were submitted to the next stages of docking.

Furthermore, for the druggability analyses, the final lists of drug target proteins were subjected to DoGSiteScorer. The DoGSiteScorer is a web-based automated pocket detection and analysis tool for calculating the druggability of protein cavities. For each detected cavity, the tool returns the pocket residues and a druggability score ranging from 0 to 1. Values closer to 1 indicate highly druggable protein cavities, i.e. the predicted cavities are likely to bind ligands with high affinity [39]. The druggable cavity for each target with value greater than 0.8 was used for the docking analysis.

## 2.5. Ligand library preparation and docking analysis

The ligand library of ZINC drug-like molecules (Natural Product and its derivatives) was downloaded from the ZINC database [40,41]. The 5008 ligands obtained in .SDF format were then converted into .PDB by using the OpenBabel (v. 2.4.1) tool [42]. After converting the file into .PDB format, the Gasteiger atomic partial charges were assigned to convert all the ligand compounds to the PDBQT format by using the prepare_ligand4.py script on the terminal. Furthermore, for the docking analysis, the final identified drug target proteins' 3D structures were examined and converted to the required PDBQT format using the AutoDockTools MGL tool (v. 1.5.4) [43]. A grid box parameter for each target comprising the residues of the DoGSiteScorer [44] druggable pocket with drug score greater than 0.8 was used for virtual screening of the ligand using AutoDock Vina [45]. The top 10 ranked ligand molecules were identified by virtual screening using the Python script topmolecule.py. Furthermore, the flexible docking was performed with the identified top 10 molecules, keeping the residues obtained from DoGSiteScorer for each target. The 3D poses of docked molecules were analysed in Chimera [46], whereas Pose View was used for two-dimensional (2D) representation [47].

## 2.6. Selection of vaccine targets

In order to test the adhesion and binding capacity to major histocompatibility complex (MHC) class I and class II, all membrane, secreted and PSE protein targets of *M. pneumoniae* were submitted to the Vaxign tool, a system based on genomic features for predicting vaccine targets in the reverse vaccinology platform. In this software, we used default parameters except for subcellular localization and transmembrane helices that were already predicted by means of SurfG+. This system has tools to identify the subcellular localization of the product of the studied sequences, analyses the transmembrane helices and is able to exclude the sequences present in non-pathogenic strains. SPAAN is a program with sensitivity of 89% and specificity of 100% that evaluates the adhesion capacity of the targets, establishing a cut-off of 0.51. The prediction of MHC-I- and MHC-II-binding epitopes is performed by Vaxitope, which searches the Immune Epitope Database (IEDB) and calculates the affinity of each molecule [48].

Sequences of the 46 PSE, secreted or membrane proteins were submitted to this platform in the FASTA format for analysis of antigenic properties. This presents the option 'Dynamic Vaxign Analysis', which is configured as the desired parameter for the prediction based on the binding capacity to MCH-I and MCH-II. Thus, proteins theoretically arising from the 88 genomes with adhesion capacity greater than 0.51 were considered immunogenic and selected for further analysis [49].

The sequences of the proteins with good MHC-binding capabilities were subsequently subjected to B-cell epitope prediction analysis to verify their ability to develop humoral immune responses. For this, we used the IEDB with a threshold of 0.5. In the platform, it is possible to analyse the proteins of interest to find the main epitopes and the value of each residue [50].

## 2.7. Analysis of proteins of interest and their interactions

To understand the metabolic interactions of the proteins of interest, we used the STRING program with default parameters, which shows the specific interactions between the proteins of *M. pneumoniae* and those present in its database, allowing the pathways' functional activities to be understood in greater

detail. For each protein–protein interaction, a score is generated. These scores represent the confidence interval ranging from 0 to 1, with 1 being the highest probability of the interaction being true. In addition to the STRING platform, other platforms were also used to contribute and reinforce the identification of the functions and metabolic pathways of proteins of *M. pneumoniae* [51].

In summary, we used the Universal Protein Resource (UniProt), which is a protein sequence and annotation database [52]. Proteins that had signal peptide were directed to the secretory pathway and were identified using the SignalP program that located the cleavage sites of each signal peptide [53]. To predict transmembrane helices, we submitted the amino acid sequences of each *M. pneumoniae* protein to the TMHMM server, which predicted the topology of these proteins by the Markov method [54].

To find out whether any of the proteins had already been tested for drug targets in previous studies, DrugBank searches were performed. DrugBank is an online database that contains information about drugs, their binding targets, interactions with other drugs, and their relationships with metabolism, gene expression and protein. Potential drugs being tested in clinical trials are also found on this platform [55].

It is well known that the use of antibiotics affects the human microbiota and is associated with immunological and metabolic alterations detrimental to the normal functioning of the organism [56]. To determine whether the proteins investigated in this study are also part of the metabolism of some of the bacteria most commonly found in the intestinal microbiota, BLASTp was performed through NCBI. Each potential drug target was submitted to the platform and compared with the bacterial protein sequences of the genera *Bacillus*, *Lactobacillus* and *Streptococcus*, which are some of the major genera found in the gut [57].

### 2.8. Analysis of genomic similarities and phylogenetic reconstruction

To compare the 88 genomes studied and to understand the differences present in each strain of *M. pneumoniae* that could enable future identification of reference genomes for pathogenicity island prediction, we used the Gegenees [58] tool; this tool fragments genomes at predefined sizes and makes an alignment of all fragments against all using the tools BLASTn, tBLAST and FASTA. With the data from this alignment, a heat map is generated that demonstrates the similarity between the lines and that ranges from 0% to 100%. The results from the Gegenees software were exported in the 'Nexus' format for later phylogenetic reconstruction using the software SplitsTree4, by using the neighbour-joining method [59].

### 2.9. Prediction of genomic islands

Prediction of genomic islands was carried out in order to identify the existence of potential drug and vaccine targets within these islands. Genomic Island Prediction Software (GIPSy) was used for the prediction of genomic islands that were classified as follows: (i) pathogenicity islands, which contain virulence factors; (ii) metabolic islands, with genes related to proteins important for metabolic pathways; (iii) resistance islands, which have genes involved in the processes of resistance to antibiotics; and (iv) symbiotic islands, with genes coding for proteins that allow the symbiotic interaction of the bacterium with the host. The characteristics analysed for predicting whether a given genome region is a genomic island were: deviations in genome signature (GC content and codon usage); the presence of transposase genes, high concentrations of virulence factors, genes related to antibiotic resistance, metabolic pathways and symbioses for pathogenicity, resistance, and metabolic and symbiotic islands, respectively; the presence of insertion sequences or flanking tRNA genes; and size ranging from 6 to 200 kb [60].

The genomes used in this step were selected from the results of the phylogenetic analysis. The phylogenetically closest lineages according to the SplitsTree [56] program were organized into clusters and, from that result, 15 genomes were selected for prediction analyses of genomic islands. The genome of the species *Mycoplasma gallinarum*, which is phylogenetically close to the species *M. pneumoniae* but is not pathogenic to humans, was selected as a reference in predicting the islands. The results obtained by GIPSy were later plotted in a circular figure using the software BRIG [61,62].

## 3. Results

The key steps for target identification, the methodologies used and the total number of proteins described in each step are summarized in the workflow of figure 1.

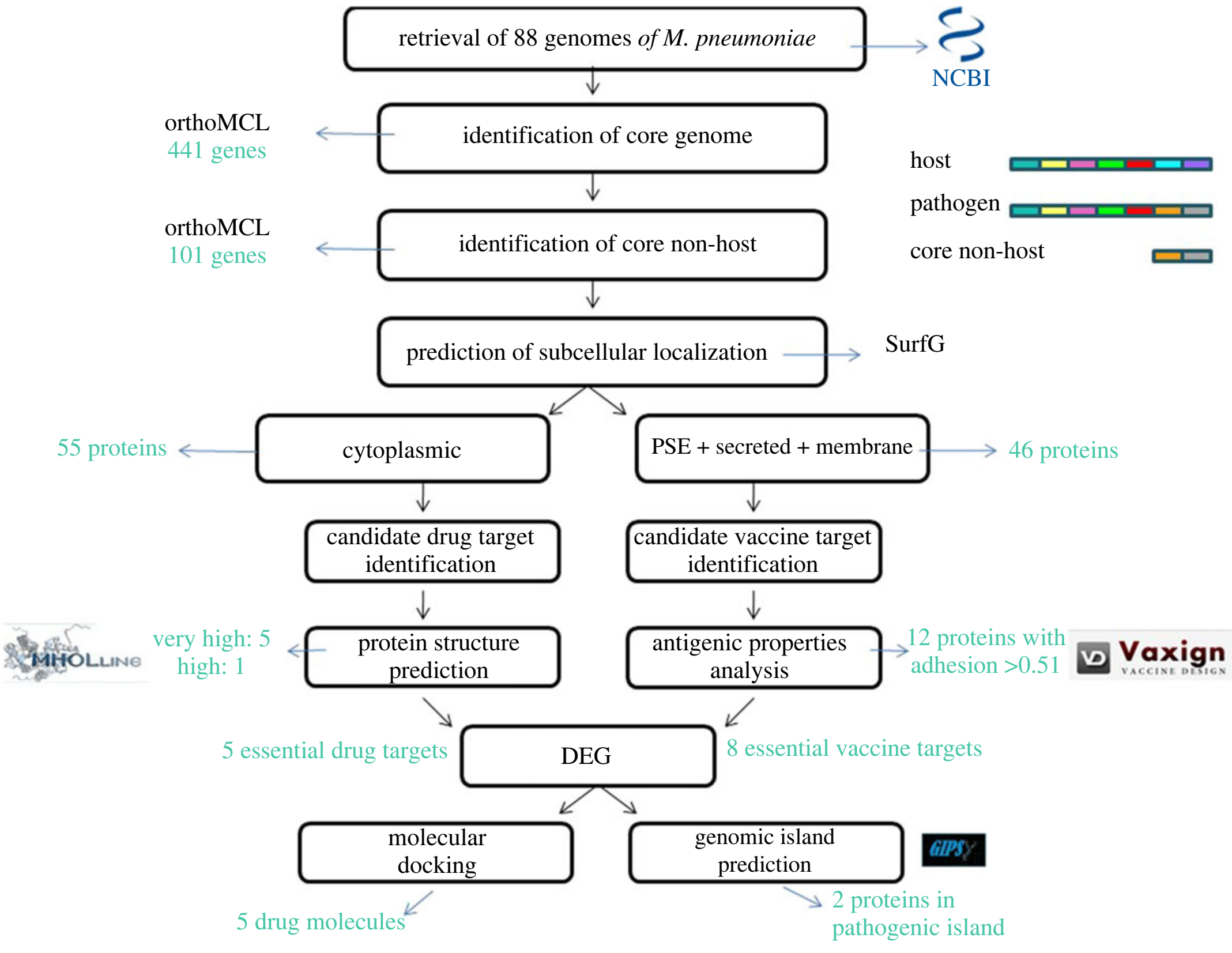


**Figure 1.** Workflow projected with the methodologies used and the total number of proteins identified in each step.

## 3.1. Identification of *M. pneumoniae* conserved proteins and subtractive genomics

Using the software OrthoFinder, we found 441 genes belonging to the core, 289 shared genes and 50 singletons. After the subtractive genomic analysis with these core genes, the number was reduced from 441 to only 101 potential targets.

## 3.2. Localization of target proteins

As described previously, for the prediction of protein localization we employed the software SurfG+. From the 101 targets, 55 proteins were predicted as cytoplasmic and directed to drug targeting. The other 46 proteins, considered PSE, secreted or from membrane were directed to analyses for vaccine targets (table 1).

## 3.3. Drug target identification and druggability analysis

The proteins predicted as cytoplasmic and essential for the bacteria are frequently considered good candidates for drug targets [35]. Thus, the protein sequences predicted as cytoplasmic were submitted to MHOLline, which used the HMMTOP, BLAST, BATS, MODELLER and PROCHECK software to predict 3D modelling. Based on the drug target analysis, only those proteins from the G2 group ($E \leq 10 \times 10^{-5}$, identity $\geq 25\%$ and LVI $\leq 0.7$) were selected. Five proteins with very high classification were identified within the program criteria and one with high potential (electronic supplementary material, file S1). The other proteins with lower level of quality were discarded. These six proteins were also submitted to the DEG and only five of them were considered vital for *M. pneumoniae*, following the criteria of bit score of 100 and $E$-value with a cut-off of $1 \times 10^{-4}$ (table 2).

The five proteins identified as potential candidates for drug action were ribosome-binding factor A (WP_010874513.1), division/cell-wall cluster transcriptional repressor MraZ (WP_010874670.1), dTIGR00282 family metallophosphoesterase (WP_010874705.1) and the hypothetical proteins WP_010874779.1 and WP_014325598.1.

Ribosome-binding factor A (WP_010874513.1) is one of the most important bacterial proteins that assist in the late stages of maturation of the 30S ribosomal subunit. Furthermore, such a protein is

**Table 1.** Localization of target proteins.

| location | number of proteins |
|---|---|
| cytoplasmic | 55 |
| PSE | 15 |
| secreted | 3 |
| membrane | 28 |
| total | 101 |

essential for the efficient processing of 16S rRNA and may interact with the 5′-terminal helix region of 16S rRNA (116 aa). Its metabolic interactions occur through interactions with the following proteins: (i) translation initiation factor (IF); (ii) phenylalanine–tRNA ligase; (iii) translation elongation factor; and (iv) ribosomal protein S15. Ribosomal protein S15 is one of the major rRNA-binding proteins that binds directly to the 16S rRNA, where it assists in the assembly of the 30S subunit platform by ligating and joining several helices of 16S rRNA [63]. Aminoglycosides are antibiotics that bind to the 30S ribosomal subunit, causing base modifications that consequently modify codon reading by interfering with mRNA translation [64]. Thus, although no ribosome-binding factor A (WP_010874513.1) studies have been found as a potential drug target, it is believed that interfering with its binding to the 30S subunit may interfere with its structural function and lead to transcriptional errors that may affect bacterial protein synthesis (electronic supplementary material, files S2 and S3).

Division/cell-wall cluster transcriptional repressor MraZ (WP_010874670.1) is a DNA-binding transcription factor, which interacts with (i) HrcA transcription repressor, which is a negative regulator of class I heat shock genes (operons grpE-dnaK-dnaJ and groELS); (ii) protein B of segregation and condensation, which acts on the chromosomes during cell division; (iii) IF-3, which binds to the 30S ribosomal subunit, increasing the availability of those subunits in which the initiation of the protein synthesis begins; (iv) chaperone proteins that prevent aggregation of stress-depleted proteins; and (v) protein RecA, which can catalyse the hydrolysis of ATP in the presence of single-stranded DNA (electronic supplementary material, files S4 and S5).

WP_010874705.1 is a protein of the metallophosphoesterase family, which is related to DNA repair (electronic supplementary material, files S6 and S7) [65]. WP_010874779.1 is a hypothetical protein whose function remains incompletely elucidated, but, according to STRING's predictions, it participates in interactions with chromosomal segregation proteins, carrier proteins and endonucleases (electronic supplementary material, files S7 and S9). Finally, WP_014325598.1 is also a hypothetical protein. It is related not only to the folding and transport of proteins, but also to tRNA ligase of threonine and arginine in addition to the tRNA responsible for thiamine synthesis (electronic supplementary material, files S10 and S11).

When comparing these proteins with the proteome of a group of bacteria present in the intestinal microbiota (*Bacillus/Lactobacillus/Streptococcus* group) through BLAST NCBI, we observed that three of the five potential drug targets present a protein profile similar to those of the database. 30S ribosome-binding factor showed 28% identity with the protein 30S ribosome-binding factor RbfA, which is present in *Lactobacillus sanfranciscensis*, and 24% identity with the same protein in *Lactobacillus pantheris*. The transcriptional regulator MraZ showed identity of about 40% with a series of *Bacillus* species, a result similar to that found through the BLAST analysis of the protein dTIGR00282 from the metallophosphoesterase family.

## 3.4. Molecular docking and virtual screening

Natural products have played important roles in recent drug development, where an enormous number of natural product-derived compounds in various stages of clinical development were highlighted [66]. For each target protein, 5008 drug-like compounds (Natural Product and its derivatives) were screened from the ZINC database. The top 10 compounds obtained by means of the AutoDock Vina binding affinity score (electronic supplementary material, file S12) were further used for flexible docking analysis with the residues of the most druggable cavity identified by DoGSiteScorer (table 4). As a result, the predicted protein–ligand interactions for best ligands with each target are displayed in

**Table 2.** Drug targets.

| target | ID | name | gene UniProt | length (aa) | molecular weight (Da) UniProt | structural quality MHOLline | biological process |
|---|---|---|---|---|---|---|---|
| 1 | WP_010874513.1 | ribosome-binding factor A | rbfA | 116 | 13 389 | very high | maturation of the functional nucleus of the 30S ribosomal subunit |
| 2 | WP_010874670.1 | transcriptional regulator MraZ | MraZ | 141 | 16 335 | very high | division/cell-wall cluster transcriptional repressor MraZ |
| 3 | WP_010874705.1 | dTIGR00282 family metallophosphoesterase | MPNE_0406 | 281 | 31 431 | very high | metal ion binding |
| 4 | WP_010874779.1 | hypothetical protein MPN423 | MPN_423 | 129 | 14 939 | very high | hydrolase activity, metal ion binding |
| 5 | WP_014325598.1 | hypothetical protein | MPN_555 | 193 | 22 434 | very high | protein folding protein transport |

**Table 3.** Identified druggable pocket with its volume, surface area and drug score of each target protein obtained from DoGSiteScorer.

| protein name | volume (Å$^3$) | surface area (Å$^2$) | drug score | residues |
|---|---|---|---|---|
| 30S ribosome-binding factor (WP_010874513.1) | 1125.38 | 1672.38 | 0.82 | TYR1, LYS5, LYS6, GLU7, ARG8, LEU9, GLU10, ASN11, ASP12, ILE13, ILE14, LEU16, ILE17, ASN18, VAL21, VAL30, LYS31, THR32, GLY33, HIS34, VAL35, THR36, HIS37, VAL38, LYS39, LEU40, ASP42, ASP43, LEU44, VAL47, VAL49, LEU51, VAL63, PHE66, ASN67, ALA69, LYS70, PHE73, VAL76, LEU77, ASN80, ILE89, HIS90, PHE91 |
| division/cell-wall cluster transcriptional repressor MraZ (WP_010874670.1) | 395.39 | 672.27 | 0.76 | ASN33, ARG34, GLY35, PHE36, GLU37, ASN38, CYS39, LEU40, GLU41, TYR51, LEU68, LEU71, ILE72, ASP72, ASP96, ALA97, ILE106, GLN108, HIS111, GLU113, TRP115, TYR120, TYR123, LEU124 |
| dTIGR00282 family metallophosphoesterase (WP_010874705.1) | 177.28 | 311.54 | 0.31 | LYS49, ASN71, HIS72, TRP74, PHE75, PHE99, LEU130, PRO131, PHE132 |
| hypothetical protein (WP_010874779.1) | 423.81 | 585.63 | 0.66 | PHE62, SER66, VAL69, VAL86, LYS87, CYS89, CYS90, PHE93, TYR94, LEU97, PHE100, ILE101, LEU104, TYR115, LEU119, GLY120, PHE123, GLY124, VAL125 |
| hypothetical protein (WP_014325598.1) | 568.26 | 839.56 | 0.81 | LYS45, GLU130, ILE131, THR132, VAL135, VAL139, ILE140, TYR143, TYR144, GLU145, THR147, ASN148, TYR154, VAL164, ALA167, LEU168, GLU171, ARG172, LEU175 |

**Table 4.** Docking studies of drug-like molecules (compounds) from the ZINC database with five drug target proteins. The table shows the binding scores/affinity, number of hydrogen bonds and the residues of proteins interacting with the respective compounds.

| ZINC compound ID | AutoDock Vina binding affinity | no. of H-bond/residues |
|---|---|---|
| 30S ribosome-binding factor (WP_010874513.1) | | |
| ZINC04259381 | −10.5 | 3/ASN18, ARG15 |
| division/cell-wall cluster transcriptional repressor MraZ (WP_010874670.1) | | |
| ZINC04235924 | −10.2 | 1/ARG34 |
| dTIGR00282 family metallophosphoesterase (WP_010874705.1) | | |
| ZINC04259703 | −8.9 | 3/LYS49, ASN71 |
| hypothetical protein (WP_010874779.1) | | |
| ZINC05415832 | −11.1 | 1/PHE93 |
| hypothetical protein (WP_014325598.1) | | |
| ZINC04236030 | −10.3 | 2/LYS45, TYR154 |

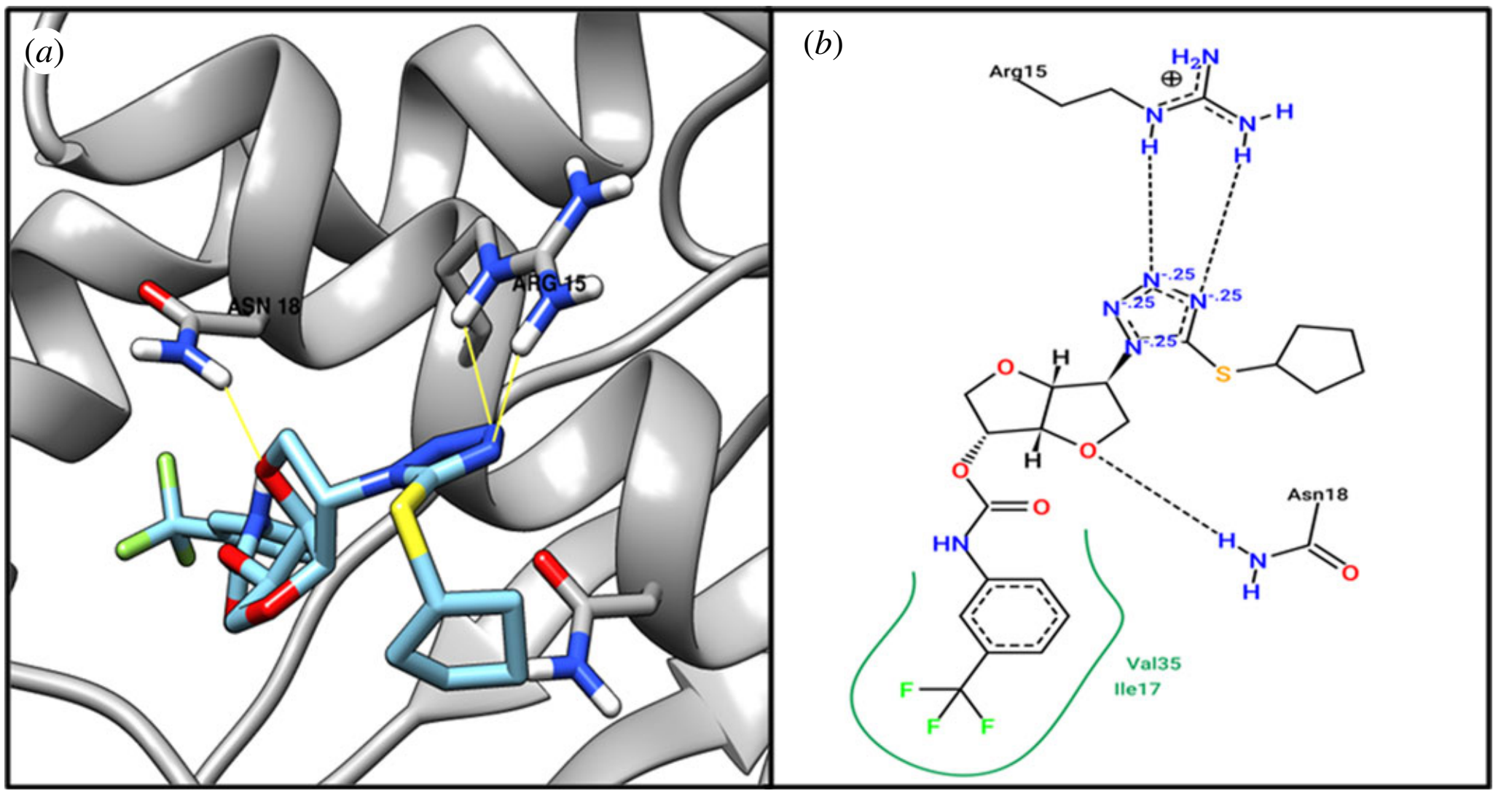


**Figure 2.** The (*a*) 3D and (*b*) 2D representation docking analyses of protein 30S ribosome-binding factor (WP_010874513.1) with compound ZINC04259381.

table 4, with ZINC database compound ID, AutoDock Vina binding affinity for the selected ligands as well as interactions of hydrogen bonds with the targets' residues involved in the interaction.

Based on structural comparison with a crystallographic structure of 30S ribosome-binding factor (WP_010874513.1) template (PDB ID: 1pa4) (putative ribosomal protein), we performed active site identification analysis with DoGSiteScorer [44], an online tool for active site residues (table 3). By performing virtual screening of 5008 drug-like molecules, we identified the top 10 molecules (electronic supplementary material, file S12); then flexible docking was performed on the identified top 10 molecules to find interactions with residues of the 30S ribosome-binding factor (WP_010874513.1) protein. We found that compound ZINC04259381 interacts with the active residue ASN18 from our active site identification analysis (table 4). Figure 2 shows the 3D and 2D representations of compound ZINC04259381.

Target MraZ (cell-wall cluster transcriptional repressor) protein, which is a transcription factor of *Escherichia coli*, regulates its own operon, also known as the division and cell wall (DCW) cluster [67]; active residue ARG34 evidenced an interaction with compound ZINC04235924 (table 4), while figure 3 shows the 3D and 2D representations of compound ZINC04235924. The target dTIGR00282 family metallophosphoesterase showed interaction with compound ZINC04259703 and interacts with the active residues LYS49 and ASN71 (table 4). Figure 4 shows the 3D and 2D representations of compound ZINC04259703. The hypothetical protein WP_010874779.1 showed interaction with compound ZINC05415832, by interacting with residue PHE93 (table 4). Figure 5 displays the 3D and 2D molecular representations of compound ZINC05415832. The hypothetical protein WP_014325598 evidenced interaction with compound ZINC04236030, by interacting with residues LYS45 and TYR154. Figure 6 depicts the 3D and 2D molecular representations of compound ZINC04236030.

## 3.5. Vaccine targets

From the 46 proteins predicted as membrane, PSE or secreted and whose structures were evaluated positively for adhesion capacity to MHC-I and MHC-II, 12 were noted with probability higher than 0.51 and considered good targets. They were also submitted to the DEG database, which indicated that eight of them were considered essential for *M. pneumoniae*. Among the eight potential vaccine targets found through these analyses, seven were lipoproteins and three of these belong to a specific group of membrane proteins of *M. pneumoniae*, with a lipid binding site and characterized as a membrane anchor (table 5).

The pro-lipoprotein diacylglyceryl transferase (WP_010874581.1) is an enzyme that catalyses the first step in the biogenesis of lipoproteins. It transfers the *n*-acyl diglyceride group into an N-terminal cysteine of the membrane lipoproteins. It is also an integral membrane protein that participates in a number of interactions; for example, the signal peptidase protein II that catalyses the removal of peptides that

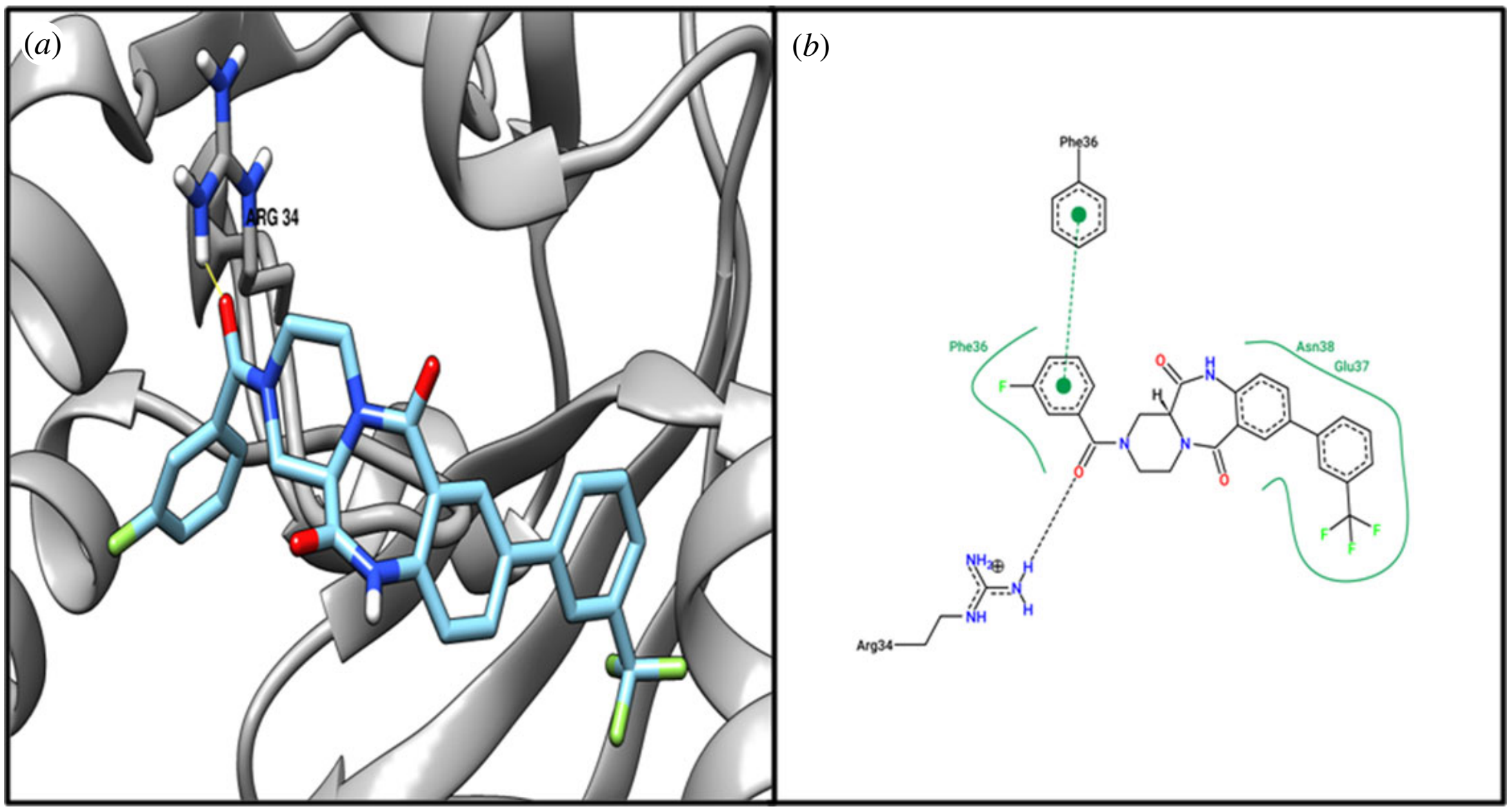


**Figure 3.** The (*a*) 3D and (*b*) 2D representation docking analyses of protein division/cell-wall cluster transcriptional repressor MraZ (WP_010874670.1) with compound ZINC04235924.

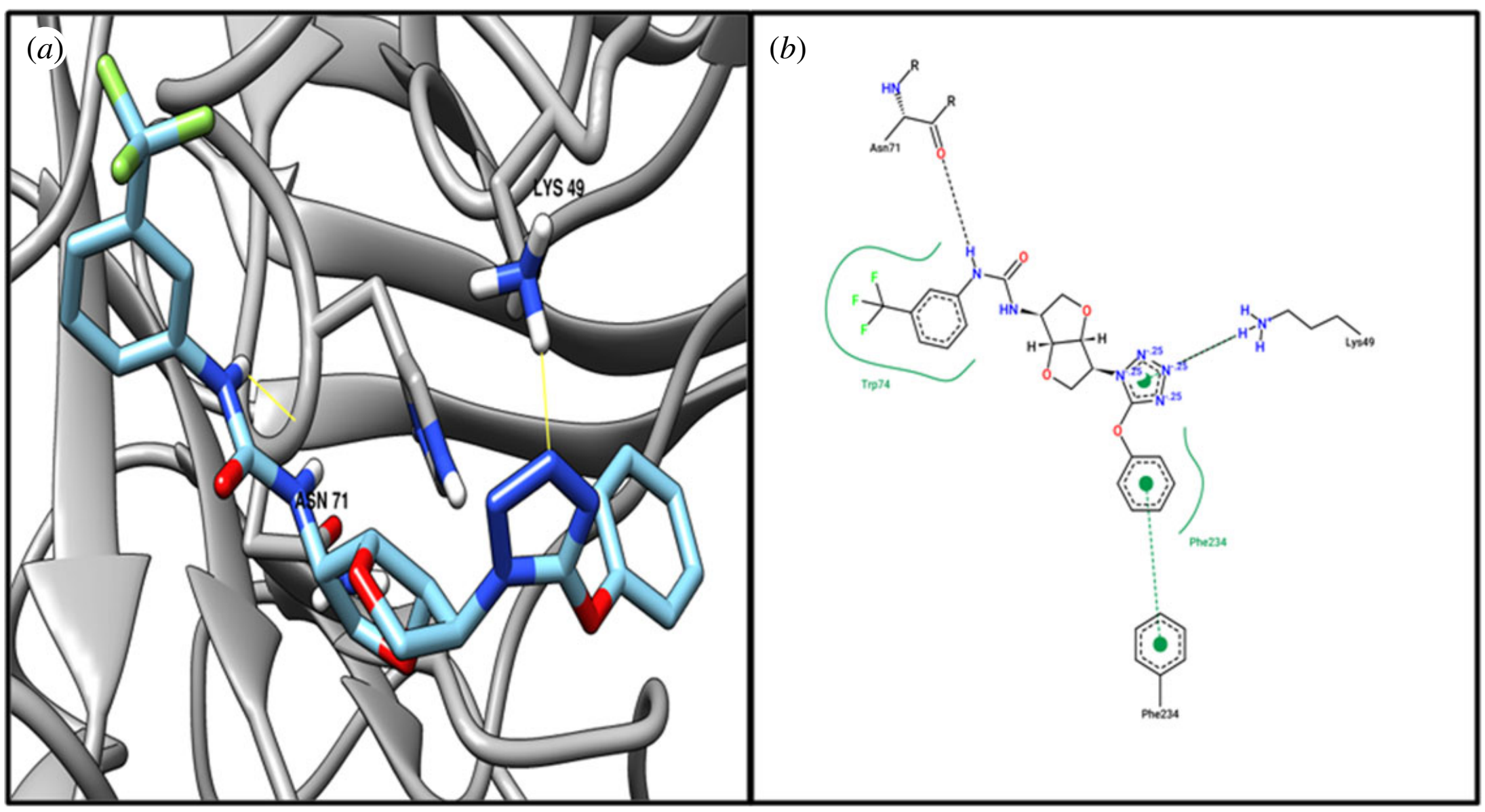


**Figure 4.** The (*a*) 3D and (*b*) 2D representation docking analyses of protein dTIGR00282 family metallophosphoesterase (WP_010874705.1) with compound ZINC04259703.

signal pro-lipoproteins. In addition, it is also involved with proteins that present DNA repair properties. The targets found by Vaxign with better MHC adhesion capacity were the proteins WP_014325486.1, with an adhesion index of 0.618, and WP_010874862.1, with index of 0.667. These two lipoproteins, which are predicted to belong to the cytoplasmic membrane or exposed to the surface, include 793 amino acids in their composition. The WP_010874581.1 protein showed seven transmembrane domains while the WP_014325660.1 protein showed one domain. The other proteins displayed no predicted domains through TMHMM (table 5).

The candidate proteins for vaccine targets were subjected to the antigenic prediction of B-cell epitopes. For each protein, the number of peptides with ability to induce the humoral immune response was predicted. We found 19 epitopes on the WP_010874862.1 protein and 16 epitopes on the WP_014574866.1 protein. Epitopes with fewer than seven amino acids were discarded from the study because they are considered too small to induce immunogenicity (electronic supplementary material, files S13–S20).

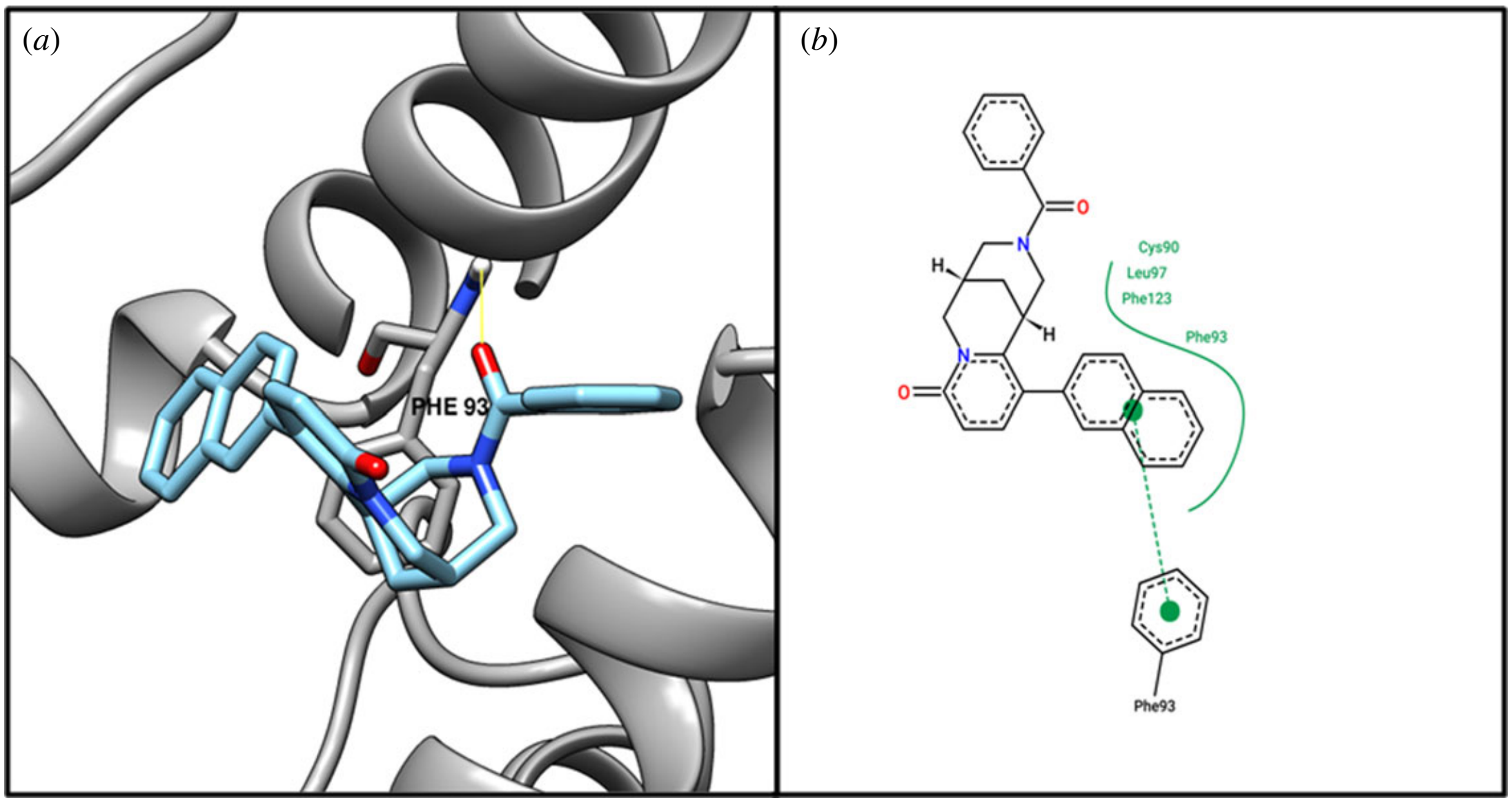


**Figure 5.** The (*a*) 3D and (*b*) 2D representation docking analyses of hypothetical protein (WP_010874779.1) with compound ZINC05415832.

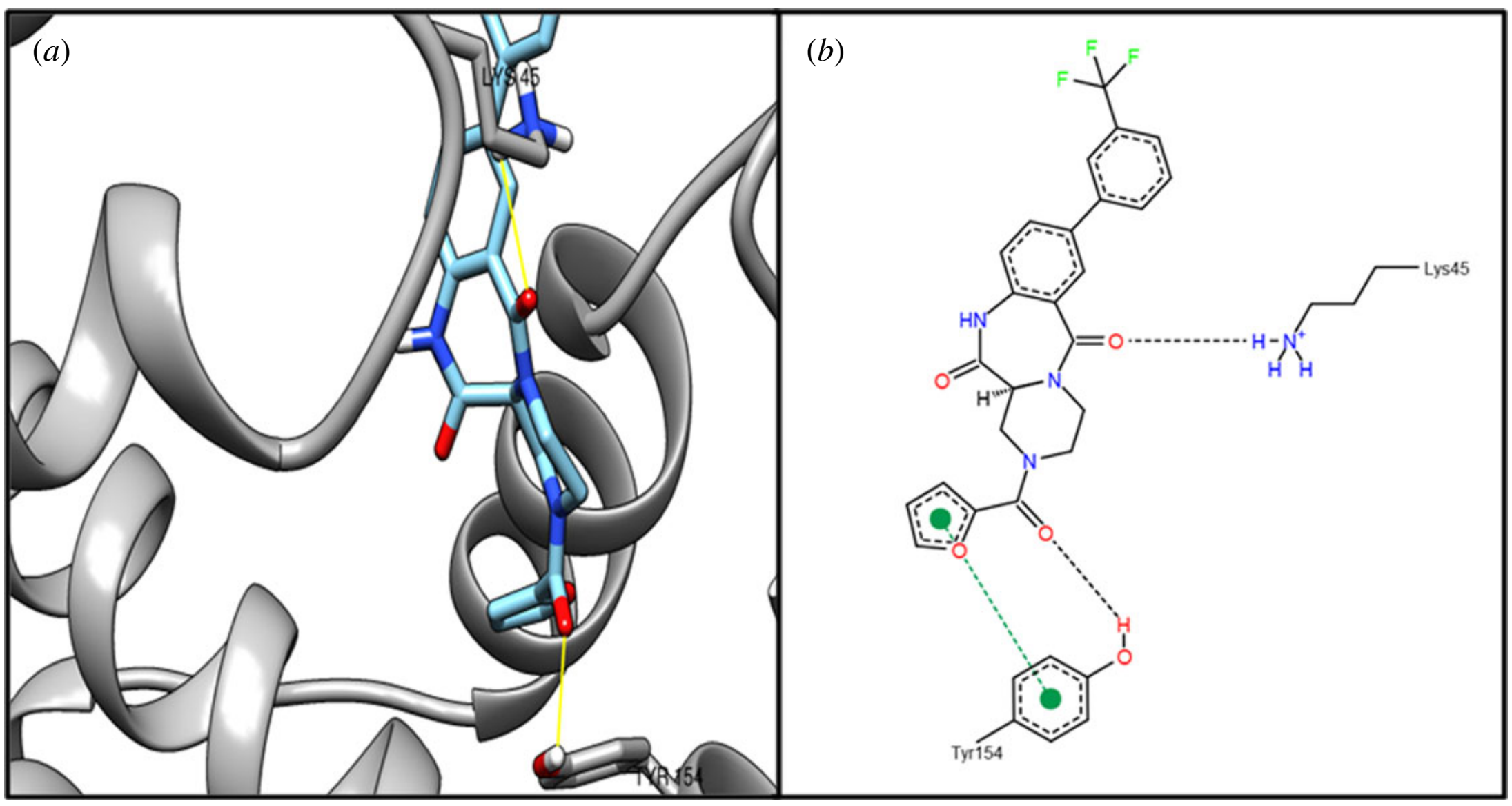


**Figure 6.** The (*a*) 3D and (*b*) 2D representation docking analyses of hypothetical protein (WP_014325598.1) with compound ZINC04236030.

## 3.6. Analysis of genomic similarities and phylogenetic reconstruction

The 88 genomes studied showed a high similarity. The heat map generated by the software Gegenees presented colours ranging from green (high similarity) to red (low similarity). Most genomes presented approximately 99% similarity, with the lowest being 95%. In the phylogenetic reconstruction performed through the software SplitsTree4, we can note the formation of seven clusters of *M. pneumoniae* genomes organized according to their phylogenetic characteristics (electronic supplementary material, files S21 and S22).

## 3.7. Prediction of genomic islands

In order to predict genomic islands, we selected 15 genomes belonging to the different clusters observed in the phylogenetic analyses. A genome of the species *M. gallinarum* was used as a reference for this

**Table 5.** Vaccine target candidates for *M. pneumoniae* identified by Vaxign. TMHMM, transmembrane propeller prediction software.

| target | ID | name | location SurfG+ | adhesin probability | no. predicted epitopes | TMHMM | protein length (aa) | SignalP | gene | molecular weight (DA) UniProt |
|---|---|---|---|---|---|---|---|---|---|---|
| 1 | WP_010874999.1 | *Mycoplasma* specific lipoprotein, type 3 | PSE | 0.529 | 5 | 0 | 279 | yes 25–26 | MPN_642 | 31 287 |
| 2 | WP_014574866.1 | hypothetical lipoprotein | PSE/outer membrane | 0.557 | 16 | 0 | 524 | no | MPN_084 | 59 553 |
| 3 | WP_010874581.1 | pro-lipoprotein diacylglyceryl transferase | cytoplasmic membrane | 0.578 | 9 | 7 | 389 | no | MPN_XXX (lgt) | 44 596 |
| 4 | WP_010874862.1 | uncharacterized lipoprotein MPN_506 | PSE/cytoplasmic membrane | 0.618 | 19 | 0 | 793 | yes 24–25 | MPN_506 | 87 494 |
| 5 | WP_014325486.1 | uncharacterized protein | PSE/outer membrane | 0.667 | 16 | 0 | 793 | yes 24–25 | MPNE_0422 | 87 951 |
| 6 | WP_014325517.1 | uncharacterized lipoprotein MPN_408 | PSE | 0.606 | 15 | 0 | 760 | yes 28–29 | MPN_408 | 83 344 |
| 7 | WP_014325659.1 | uncharacterized lipoprotein MG440 | PSE | 0.536 | 7 | 0 | 277 | yes 26–27 | MPN_646 | 31 097 |
| 8 | WP_014325660.1 | uncharacterized lipoprotein MG439 homologue 1 | extracellular | 0.543 | 5 | 1 | 290 | yes 28–29 | MPN_647 | 31 823 |

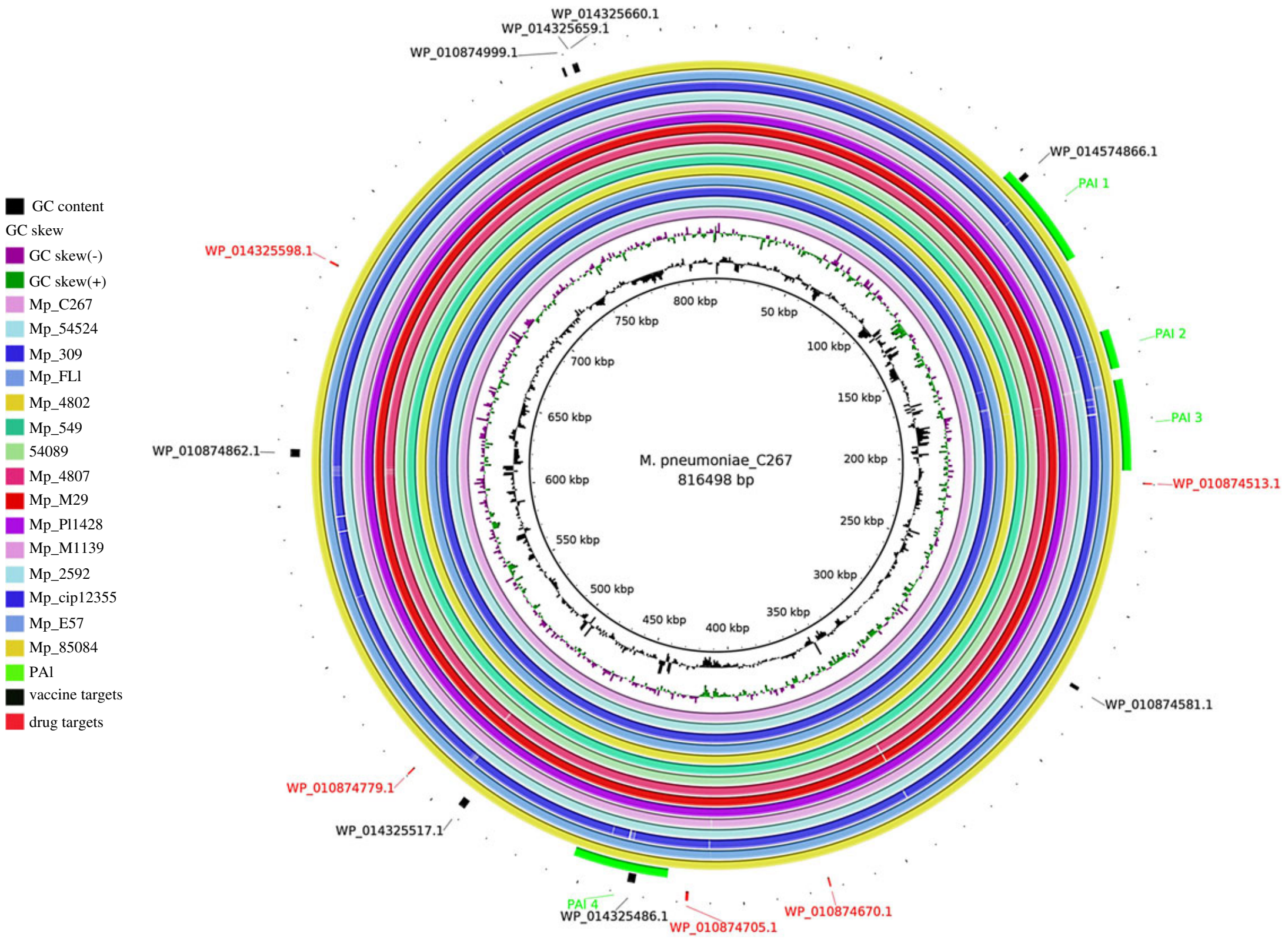


**Figure 7.** Pathogenicity islands predicted using 15 strains of *M. pneumoniae* having *M. pneumoniae* C267 as reference and *M. Gallinarum* as non-pathogenic reference.

prediction, given that this species is not pathogenic for humans. Four pathogenicity islands common to 15 genomes were predicted. These islands were common to all strains of *M. pneumoniae* tested and are very similar, as seen in the image generated by BRIG. For example, PAI1 is present in all strains and exhibits minimal deletions only in the CIP12355 and MP4807 strains. Among the vaccine targets, the WP_014574866.1 protein was predicted within PAI1 and the WP_014325486.1 protein within PAI4. The other proteins indicated as vaccine targets were not found in regions of pathogenicity islands. No potential drug targets were predicted among the genomic islands (figure 7).

# 4. Discussion

*Mycoplasma pneumoniae* is the main pathogen for pneumonia in children and, while the diagnosis is often limited, its incidence and number of antimicrobial-resistant cases have increased worldwide [68,69]. The scenario gets even worse if we consider the persistent absence of prophylactic methods and the fact that there are few treatment options for acute and chronic infections by this pathogen [18,70]. In the present study, comparative genomics and reverse vaccinology of 88 *M. pneumoniae* genomes were carried out in an attempt to predict vaccine and drug targets that could be tested in the near future in order to solve this public health problem. In the present study, employing the software OrthoFinder, 441 proteins belonging to the core genome were found and these targets were common to all 88 strains analysed throughout all coding sequences. Through subtractive genomics, we identified the proteins homologous to human proteins and removed them from the study, so that the selected targets act only against *M. pneumoniae*, preventing possible adverse reactions. After this filtration, 101 proteins remained. In order to evaluate the 101 proteins for their usefulness as drug targets or vaccines, we defined the subcellular location as the main parameter. Of the 101 proteins, 44 proteins considered membrane, PSE or secreted were, therefore, selected for the analysis of vaccine targets. The other 55 were analysed for their ability to act as drug targets. Finally, using reverse vaccinology, we found seven proteins with high potential for vaccine use and five with great potential for drug targeting.

The immune system can identify molecules foreign to our body and generate an immune response acquired through the interaction of these molecules with the MHC present in antigen-presenting cells

such as dendritic cells and macrophages. At the MHC-binding site, these foreign antigens are presented to $CD4^+$ and $CD8^+$ T cells in a process called antigen presentation, a process essential for the activation of the adaptive immune response and generation of the differential pattern of $CD4^+$ and $CD8^+$ T-cell immunity [71,72]. B cells are also involved in this immune response because they are also important antigen-presenting cells whose presentation is essential for the development of humoral immunity. The potential antigenic targets found in this study have not yet been tested *in vivo* for their ability to induce one or another pattern of immune response. Thus, further studies must be performed in order to test these targets and ascertain whether the best humoral and/or cellular immune response pattern is induced [73,74].

In the present study, 12 proteins were found capable of adhering to MHC-I and MHC-II with an index greater than 0.51, which means that they may induce either cellular or humeral adaptive immune responses [75]. As mentioned previously, these proteins are found either in the membrane, PSE or secreted by *M. pneumoniae* and, therefore, are the first to come into contact with host defences, given that *M. pneumoniae* has no cell wall and its location leaves them more exposed to the extracellular environment, which in turn facilitates recognition and specific memory immune responses [76,77]. Eight of the 12 proteins tested for *M. pneumoniae* essentiality were found to be strictly essential according to the software DEG.

From the amino acid sequences of these proteins, we predicted their B-cell epitopes. These epitopes can be recognized by the immune system, thus contributing to the development of immunity through the production of antibodies. Thus, in the present study, all eight proteins presented regions with B-cell interaction capabilities and could be used as vaccine targets [50].

Of these, only two proteins showed transmembrane domains. WP_010874581.1, belonging to the cytoplasmic membrane, presented seven transmembrane helices while the PSE protein WP_014325660.1 presented a predicted domain. These helices cross the outer membrane several times, forming loops where the epitopes are exposed, with precisely organized amino acids, thus enabling contact with the immune system [78]. However, proteins with more than one transmembrane helix in their structure hamper purification in assays for vaccine production [79].

Cytoplasmic proteins act in the maintenance of cell survival, and, for this reason, all 55 sequences of the 55 cytoplasmic proteins found were analysed for their potentials as drug targets. For this, the MHOLline tool was used, which identifies the set of 3D models of proteins on the core non-host. We found six proteins with $E \leq 10 \times 10^{-5}$, identity $\geq 25\%$ and LVI $\leq 0.7$, criteria used to verify the significance in modelling. We prioritized the five proteins considered essential by the software DEG for further study because, if the target interferes with some vital metabolic pathway of the bacteria, the effectiveness of the possible drug that will come into contact with this protein will probably be greater.

The first protein found with potential as a drug target was WP_010874513.1 (ribosome-binding factor A) and this molecule is essential in the processing of 16S rRNA. The protein WP_010874670.1 (division / cell-wall cluster transcriptional repressor MraZ) interacts with components of cell division; this aspect when analysed with regard to drug targets is interesting since it is a vital cellular process. The third predicted protein, WP_010874705.1, from the metallophosphoesterase family, may also be considered a potential target since it relates to DNA repair, another very important process for cell integrity. The functions of the other two proteins found as potential drug targets remain incompletely elucidated. WP_010874779.1 is a hypothetical protein that according to STRING's predictions interacts with endonucleases, carrier proteins and proteins of the chromosomes, a property that may be theoretically important. The last protein, WP_014325598.1, is also a hypothetical protein related to the folding and transport of protein. These activities are intrinsically related to many important functions of any cell and, therefore, changes in these activities may compromise all cell biology.

The software AutoDock Vina was used for docking analysis. The five protein targets 30S ribosome-binding factor (WP_010874513.1), division/cell-wall cluster transcriptional repressor MraZ (WP_010874670.1), dTIGR00282 family metallophosphoesterase (WP_010874705.1), hypothetical protein (WP_010874779.1) and hypothetical protein (WP_014325598.1) were tested for their efficacy in binding to natural compounds obtained from the ZINC database that can act as drugs. Lower levels of energy and other parameters are related to greater interaction capacities [80]; so, we found 50 ligands with high druggability and, for each protein target, one of those ligands was selected to verify the structural interaction. ZINC04259381, ZINC04235924, ZINC04259703, ZINC05415832 and ZINC04236030 are the identified compounds with high affinity to bind the proteins. ZINC04259381 is the molecule with the best affinity score and binds to 30S ribosome-binding factor (WP_010874513.1), a target that is involved with RNA processing; and any alteration in this pathway can lead to cell

death. Therefore, ZINC04259381 is identified as the best drug candidate in our analysis and both identified molecules could be considered a candidate for antimicrobial chemotherapy in future studies for the development of drugs against pneumonia caused by *M. pneumoniae*.

The phylogenomic analysis was performed through two software programs, Gegenees and SplitsTree. This evaluation demonstrated the relationships and differences between the strains and modifications that normally occur during the evolution of microorganisms, including bacteria. These data reveal that, despite the differences between the strains of *M. pneumoniae*, visible in the phylogenetic tree generated by SplitsTree, the genomes were very similar. This proved to be very important given that the main goal of this work was to find potential targets for drugs and vaccines that could act against all strains of *M. pneumoniae*. This high level of genetic similarity is seen as a result of a degenerative evolution process, in which losses of genomic regions occurred over time, leaving only those genes essential for the species [81,82].

To understand the relationship of *M. pneumoniae* with eukaryotic cells, as well as their evolution, the studies on pathogenicity islands and virulence factors were essential. This information has already been shown to be important for the development of new methods of treatment and vaccination against bacteria [83]. From the methodology, we found four pathogenicity islands present in all 15 strains used. Comparison of the genomes performed via the software BRIG showed that there are few regions of deletion between the genomes, which indicates a small difference between the islands. This fact suggests that these islands already existed in the ancestral species that gave rise to *M. pneumoniae*.

Of all the targets that were detected in this research study, only the proteins WP_014574866.1 and WP_014325486.1 were found on pathogenicity islands. In this reverse vaccinology approach, previous studies reported that proteins associated with pathogenicity islands are considered to be excellent vaccine targets [22]. The protein WP_014325486.1 is PSE. This protein does not have a well-elucidated structure or functions, but it does have the highest capacity to bind to MCH-I and MHC-II, according to the results generated through Vaxign; it was also the protein with the highest number of predicted epitopes capable of activating B cells and developing the humoral immune response. All these characteristics reinforce the inference that this may be a good candidate for vaccines. Therefore, we believe that *in vivo* experiments should follow this direction and these targets should be tested in the near future.

# 5. Conclusion

CAP causes millions of deaths worldwide, an outcome that could be avoided with the development of appropriate prophylactic and treatment methods. In the present study, 88 genomes deposited in the NCBI database were employed to predict *in silico* proteins that can be used as vaccine targets or targets for new drugs. Through reverse vaccinology and subtractive genomic approaches, seven proteins with potential to induce immune responses were predicted as vaccine targets for protection against different strains of *M. pneumoniae*, a bacterium responsible for most of the infections that lead to pneumonia. Since treatment for this type of infection is limited because of the bacterium's high resistance to antibiotics, the genomes were also submitted to comparative genomic analysis that identified five possible drug targets. These targets were compared with different databases through molecular docking and should be subjected to future analyses as potential therapeutic resources.

Taking all the data together, we can assert that the current work presents great relevance to world health for finding new therapeutic targets for pneumonia due to *M. pneumoniae* infection. These targets could be quickly tested on new vaccine formulations and drug tests identified, representing a breakthrough in the area. In addition, further studies should also be performed on the other bacterial species that cause pneumonia in order to find new methods of treating the disease.

Data accessibility. The authors declare free access to the data obtained with this work. The information on how to obtain these data is indicated in the article. All the software used in the work are publicly available, as well as the genomes used for the analyses in general. Preliminary results of the gene screening have been made available; images and tables resulting from the analyses of the relationships between the lineages, molecular docking and genomic islands are also available in the electronic supplementary material along with other results obtained by the programs used during the development of the work. These files contain information on protein interactions, BLAST with the DEG and more descriptive results of epitope prediction. Access to these data can contribute to a better understanding of how it was possible to arrive at the final result and provides details on each step.

Authors' contribuitions. T.C.V.R.: carried out the download and genomes processing, carried out sequence alignments and data analyses, participated in the design of the study and drafted the manuscript. A.d.S., L.d.C.O., L.d.J.B.: carried out the download and genomes processing, carried out sequence alignments and data analyses, participated in the design

of the study. A.K.J., S.T., F.M.M.: participated in molecular docking analyses. C.J.F.O., P.G., V.A.d.C.A.: participated in the review of the article, contributing with suggestions and criticisms for approval. S.d.C.S.: conceived of the study, designed the study and coordinated the study.

Competing interests. We declare we have no competing interests.

Funding. This work was supported by the Fundação de Amparo à Pesquisa do Estado de Minas Gerais (FAPEMIG, grant no. APQ-01323-15), Coordenação de Aperfeiçoamento de Pessoal de Nível Superior (CAPES) and Conselho Nacional de Desenvolvimento Científico e Tecnológico (CNPq).

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
