## [Reviewer comments · Royal Society Open Science]

Review History

RSOS-190907.R0 (Original submission)

Review form: Reviewer 1

Is the manuscript scientifically sound in its present form?

Yes

Are the interpretations and conclusions justified by the results?

Yes

Is the language acceptable?

Yes

Is it clear how to access all supporting data?

Yes

Reports © 2019 The Reviewers; Decision Letters © 2019 The Reviewers and Editors; Responses © 2019 The Reviewers, Editors and Authors. Published by the Royal Society under the terms of the Creative Commons Attribution License <http://creativecommons.org/licenses/by/4.0/>, which permits unrestricted use, provided the original author and source are credited

Do you have any ethical concerns with this paper?

No

Have you any concerns about statistical analyses in this paper?

No

Recommendation?

Accept as is

Comments to the Author(s)

Authors implemented the suggestions provided by the reviewers.

Review form: Reviewer 2 (Asif Ullah Khan)

Is the manuscript scientifically sound in its present form?

Yes

Are the interpretations and conclusions justified by the results?

Yes

Is the language acceptable?

Yes

Is it clear how to access all supporting data?

Not Applicable

Do you have any ethical concerns with this paper?

No

Have you any concerns about statistical analyses in this paper?

No

Recommendation?

Accept with minor revision (please list in comments)

Comments to the Author(s)

The authors though addressed most of my previous comments & have considerably improved the manuscript draft. However few minor comments are still not addressed.

1. For example, the order of figures citation in the main text is not corrected. The citation of figure 8 appeared first in the main text. The figure citation order needs to be checked carefully.

2. Secondly, there are so many figures. Some of the figures should be shown as supplementary figures instead as regular article figures.

3. The Supplementary tables citation should also be checked. Somewhere, the author mentioned these as "Supplementary tables", somewhere mentioned as "Supplementary Files" and somewhere the authors just mentioned, "supplementary material" like DEG supplementary material. This should be checked & uniform.

4. As per my review, I strongly felt that the authors still need to carefully read the manuscript contents and should improve the draft with respect to typographic and grammatical mistakes.

Decision letter (RSOS-190907.R0)

18-Jun-2019

Dear Miss Vilela Rodrigues

On behalf of the Editors, I am pleased to inform you that your Manuscript RSOS-190907 entitled "Reverse Vaccinology and Subtractive Genomics reveals new therapeutic targets against *Mycoplasma pneumoniae*, a causative agent of pneumonia" has been accepted for publication in Royal Society Open Science subject to minor revision in accordance with the referee suggestions. Please find the referees' comments at the end of this email.

The reviewers and handling editors have recommended publication, but also suggest some minor revisions to your manuscript. Note particularly the minor comments of Reviewer 2. Therefore, I invite you to respond to the comments and revise your manuscript.

- Ethics statement

- Data accessibility

<http://datadryad.org/submit?journalID=RSOS&manu=RSOS-190907>

- Competing interests

- Authors' contributions

- Acknowledgements

- Funding statement

Because the schedule for publication is very tight, it is a condition of publication that you submit the revised version of your manuscript before 27-Jun-2019. Please note that the revision deadline will expire at 00.00am on this date. If you do not think you will be able to meet this date please let me know immediately.

- 1) A text file of the manuscript (tex, txt, rtf, docx or doc), references, tables (including captions) and figure captions. Do not upload a PDF as your "Main Document";
- 2) A separate electronic file of each figure (EPS or print-quality PDF preferred (either format should be produced directly from original creation package), or original software format);
- 3) Included a 100 word media summary of your paper when requested at submission. Please ensure you have entered correct contact details (email, institution and telephone) in your user account;
- 4) Included the raw data to support the claims made in your paper. You can either include your data as electronic supplementary material or upload to a repository and include the relevant doi

within your manuscript. Make sure it is clear in your data accessibility statement how the data can be accessed;

5) All supplementary materials accompanying an accepted article will be treated as in their final form. Note that the Royal Society will neither edit nor typeset supplementary material and it will be hosted as provided. Please ensure that the supplementary material includes the paper details where possible (authors, article title, journal name).

on behalf of Dr John Dalton (Associate Editor) and Steve Brown (Subject Editor)
openscience@royalsociety.org

Associate Editor Comments to Author (Dr John Dalton):

The authors still need to address the concerns of the second reviewer who point out errors on figure citations and who suggested that some figures should be presented as supplementary figures instead as regular article figures. Typographical and grammatical errors also need to be checked.

Reviewer comments to Author:

Reviewer: 1

Comments to the Author(s)

Authors implemented the suggestions provided by the reviewers.

Reviewer: 2

Comments to the Author(s)

The authors though addressed most of my previous comments & have considerably improved the manuscript draft. However few minor comments are still not addressed.

1. For example, the order of figures citation in the main text is not corrected. The citation of figure 8 appeared first in the main text. The figure citation order needs to be checked carefully.
2. Secondly, there are so many figures. Some of the figures should be shown as supplementary figures instead as regular article figures.
3. The Supplementary tables citation should also be checked. Somewhere, the author mentioned these as "Supplementary tables", somewhere mentioned as "Supplementary Files" and somewhere the authors just mentioned, "supplementary material" like DEG supplementary material. This should be checked & uniform.
4. As per my review, I strongly felt that the authors still need to carefully read the manuscript contents and should improve the draft with respect to typographic and grammatical mistakes.

Comments to the Author from the Editorial Office:

For more information about language polishing services endorsed by the Royal Society, please follow the link below:

<https://royalsociety.org/journals/authors/language-polishing/>

Author's Response to Decision Letter for (RSOS-190907.R0)

See Appendix A.

Decision letter (RSOS-190907.R1)

04-Jul-2019

Dear Miss Vilela Rodrigues,

I am pleased to inform you that your manuscript entitled "Reverse Vaccinology and Subtractive Genomics reveal new therapeutic targets against *Mycoplasma pneumoniae*, a causative agent of pneumonia" is now accepted for publication in Royal Society Open Science.

Royal Society Open Science operates under a continuous publication model (<http://bit.ly/cpFAQ>). Your article will be published straight into the next open issue and this will be the final version of the paper. As such, it can be cited immediately by other researchers.

As the issue version of your paper will be the only version to be published I would advise you to check your proofs thoroughly as changes cannot be made once the paper is published.

on behalf of Dr John Dalton (Associate Editor) and Steve Brown (Subject Editor)
openscience@royalsociety.org

Appendix A

Referee(s)' Comments to Author:

Editor Dr. John Dalton:

The authors still need to address the concerns of the second reviewer who point out errors on figure citations and who suggested that some figures should be presented as supplementary figures instead as regular article figures. Typographical and grammatical errors also need to be checked.

Authors' response:

We agree with the editor and modify the article, correct gramatical errors, the order of citation of figures and direct some of them to supplementary material.

Reviewer 1:

Authors implemented the suggestions provided by the reviewers.

Reviewer 2:

The authors though addressed most of my previous comments & have considerably improved the manuscript draft. However few minor comments are still not addressed.

1. For example, the order of figures citation in the main text is not corrected. The citation of figure 8 appeared first in the main text. The figure citation order needs to be checked carefully.

2. Secondly, there are so many figures. Some of the figures should be shown as supplementary figures instead as regular article figures.

3. The Supplementary tables citation should also be checked. Somewhere, the author mentioned these as "Supplementary tables", somewhere mentioned as "Supplementary Files" and somewhere the authors just mentioned, "supplementary material" like DEG supplementary material. This should be checked & uniform.

4. As per my review, I strongly felt that the authors still need to carefully read the manuscript contents and should improve the draft with respect to typographic and grammatical mistakes.

We consider the reviewer's suggestions and review all article writing. We also correct the citation order of the figures, standardize the supplementary files and assign some of the figures to the supplementary material.